# Profile of *TREM2*-Derived circRNA and mRNA Variants in the Entorhinal Cortex of Alzheimer’s Disease Patients

**DOI:** 10.3390/ijms23147682

**Published:** 2022-07-12

**Authors:** Amaya Urdánoz-Casado, Javier Sánchez-Ruiz de Gordoa, Maitane Robles, Miren Roldan, María Victoria Zelaya, Idoia Blanco-Luquin, Maite Mendioroz

**Affiliations:** 1Neuroepigenetics Laboratory-Navarrabiomed, Universidad Pública de Navarra (UPNA), Navarra Institute for Health Research (IdiSNA), 31008 Pamplona, Spain; amaya.urdanoz.casado@navarra.es (A.U.-C.); jsruizdegordoa@gmail.com (J.S.-R.d.G.); maitane.robles.solano@navarra.es (M.R.); mroldana@navarra.es (M.R.); iblancol@navarra.es (I.B.-L.); 2Department of Neurology, Hospital Universitario de Navarra-IdiSNA (Navarra Institute for Health Research), 31008 Pamplona, Spain; 3Department of Pathology, Hospital Universitario de Navarra-IdiSNA (Navarra Institute for Health Research), 31008 Pamplona, Spain; mv.zelaya.huerta@navarra.es

**Keywords:** circRNAs, gene expression, regulation, epigenetics, *TREM2*, Alzheimer’s disease, DAM, microglia

## Abstract

Genetic variants in *TREM2*, a microglia-related gene, are well-known risk factors for Alzheimer’s disease (AD). Here, we report that *TREM2* originates from circular RNAs (circRNAs), a novel class of non-coding RNAs characterized by a covalent and stable closed-loop structure. First, divergent primers were designed to amplify circRNAs by RT-PCR, which were further assessed by Sanger sequencing. Then, additional primer sets were used to confirm back-splicing junctions. In addition, HMC3 cells were used to assess the microglial expression of circTREM2s. Three candidate circTREM2s were identified in control and AD human entorhinal samples. One of the circRNAs, circTREM2_1, was consistently amplified by all divergent primer sets in control and AD entorhinal cortex samples as well as in HMC3 cells. In AD cases, a moderate negative correlation (r = −0.434) was found between the global average area of Aβ deposits in the entorhinal cortex and circTREM2_1 expression level. In addition, by bioinformatics tools, a total of 16 miRNAs were predicted to join with circTREM2s. Finally, *TREM2* mRNA corresponding to four isoforms was profiled by RT-qPCR. *TREM2* mRNA levels were found elevated in entorhinal samples of AD patients with low or intermediate ABC scores compared to controls. To sum up, a novel circRNA derived from the *TREM2* gene, circTREM2_1, has been identified in the human entorhinal cortex and *TREM2* mRNA expression has been detected to increase in AD compared to controls. Unraveling the molecular genetics of the *TREM2* gene may help to better know the innate immune response in AD.

## 1. Introduction

Circular RNAs (circRNAs) are a novel class of non-coding RNA characterized by the covalent junction between the 3′ end and 5′ end, generated by the back-splicing process. From the same pre-mRNA precursor, messenger RNAs (mRNA) and circRNAs are formed by canonical splicing and alternative splicing [1,2,3,4]. In the circRNAs biogenesis, not only the spliceosome machinery and the RNA polymerase II are implicated but also other elements, such as *Alu* elements, ADAR (adenosine deaminase that acts on RNA) enzyme, or RBP (RNA-binding proteins), which participate in the process [1,5,6,7,8,9,10]. This covalent union gives circRNAs a more stable structure than linear RNAs since they are resistant to the action of RNases [1], and this characteristic, among others, makes circRNAs interesting molecules to investigate in human diseases.

Moreover, circRNAs are highly evolutionary conserved, and their expression has been observed across a wide range of body tissues [1,4,10,11]. However, it is in the brain where they are most expressed. circRNAs are enriched in neuronal tissues; in fact, it has been reported that 20% of brain genes generate circRNAs [12,13]. Interestingly, neuronal specialization is related to a high level of alternative splicing [14], and this process could also explain the huge amount of brain-specific circRNAs [10].

Nevertheless, circRNAs expression levels vary from one brain structure to another, synapsis being the place where their expression is highest [10,12,13]. Their expression is also modified along neuronal development, changing space-timely and increasing its synthesis [10]. All these data suggest that RNA circularization is crucial for neuronal function. Alterations have been described in circRNAs expression levels in different neurological diseases, such as multiple sclerosis, Parkinson’s disease, amyotrophic lateral sclerosis, and Alzheimer’s disease (AD), among others [15,16,17,18].

AD is a neurodegenerative disease that is the leading cause of age-related dementia [19]. In terms of anatomopathological changes, it is characterized by extracellular accumulation of amyloid plaques and intraneuronal tau neurofibrillary tangles, along with loss of synapses between neurons [20,21,22]. At present, AD is considered a complex disease, probably caused by the interaction between three different factors: genetic, environmental, and epigenetic [20,21,22]. To date, the most widely studied epigenetic mechanisms are DNA methylation and post-transcriptional histone modifications [23,24]. However, it has been described that non-coding RNAs, such as circRNAs, also play an important role in epigenetic regulation [25,26]. The best-known circRNA implicated in AD is CDR1as (ciRS-7), generated from the Cerebellar Degeneration Related Protein 1 (*CDR1*) gene, which has more than 70 union sites for microRNA-7 (miR-7) [4,27,28]. In AD, differences in CDR1as expression levels have been observed in the hippocampus [29,30], one of the most vulnerable brain regions to tau deposition. In addition, *APP* and *MAPT* are genes encoding the main proteins involved in AD, i.e., amyloid precursor protein and tau protein, which circRNAs are also transcribed from [31,32]. Most interestingly, Aβ peptides may be synthesized from an APP-derived circRNA, known as circAβ-a [32,33]. Recent studies in AD-affected brain regions have also found differential expression in a number of circRNAs such as circHOMER1 and circCORO1C [18,34,35].

On the other hand, microglia, the innate immune cells of the brain, are also implicated in AD pathogenesis [36,37]. Over the past few years, the role of microglia in AD pathogenesis has radically changed, gaining a more relevant position in neurodegenerative diseases. In this scenario, microglia-related genes are receiving special attention, as is the case with the *TREM2* gene.

*Trem2* (Triggering receptor expressed on myeloid cells 2) is a cell surface protein almost exclusive of microglia cells and tightly associated with AD. Some single nucleotide polymorphisms (SNPs) in this gene have been described as risk factors for AD, such as p.R47H (rs75932628) or p.R62H (rs143332484) [38,39]. Moreover, in cerebrospinal fluid (CSF) of AD patients, the soluble form of *Trem2* (sTREM2) is increased [39]. sTREM2 is thought to be originated from the action of proteases on the extracellular portion of *Trem2* (C-terminal fragment) [40]. Three distinct *TREM2* transcripts have been described as expressed in the brain, i.e., ENST00000373113, ENST00000373122, and ENST00000338469 [38,41]. The first transcript (ENST00000373113) consists of five exons representing the canonical and longest *TREM2* isoform, the second transcript (ENST00000373122) is shorter than the canonical one due to the lack of exon 5 and part of exon 4; meanwhile, the third transcript (ENST00000338469) is the result of the alternative splicing of exon 4, which encodes the transmembrane domain; therefore, sTREM2 may also be translated from the latter transcript (Appendix A). Both first and third isoforms have been reported to be elevated in AD brain tissues [38,41]. Interestingly, bioinformatics by using the Genome Browser SIB Alt-Splicing Track Settings [42] also predicts that alternative splicing may involve exons 3, 4, and 5. Indeed, Yanaizu et al. described that exon 3, under normal conditions, undergoes alternative splicing in order to regulate *TREM2* expression [43]. Additionally, it has been noted an increased expression of exons 3 and 4 depending on the degree of AD-related neurofibrillary pathology [44]. Moreover, a new variant of *TREM2* lacking exon 2 has recently been described in the human brain [45,46,47]. It is interesting to note that epigenetic mechanisms involving *TREM2* are also modulated in AD since *TREM2* DNA methylation and hydroxymethylation levels are known to be altered in the hippocampus [48] and other brain regions affected by AD [49,50,51].

Despite the increasingly relevant involvement of circRNAs in the brain and the key role of *TREM2* in neurodegenerative processes, no circRNAs originating from *TREM2* have been described in AD so far. After examining the genetic structure of *TREM2*, we hypothesize that alternative splicing of exon 4 could be favoring *TREM2*-originated circRNAs (circTREM2s), which in turn may be involved in the pathogenesis of AD. We decided to study exon 4 as this exon encodes the transmembrane domain and is alternatively spliced to form the sTREM2 isoform. Thus, we used a human embryonic microglia cell line (HMC3 cells) and human entorhinal cortex, a brain region most vulnerable to AD pathology, from AD patients and controls to identify *TREM2*-originated circRNAs. Additionally, we profiled *TREM2* mRNA expression for the four different isoforms in human entorhinal cortex samples.

## 2. Results

### 2.1. circTREM2 Identification in Entorhinal Human Cortex

So far, no circTREM2 has been described in human tissues. *TREM2* generates three isoforms by alternative splicing [38,41], and in turn, *TREM2* exons 2, 3, and 4 skipping has been shown to be involved in AD [43,44,45]. Therefore, we hypothesized that exon 4, which is skipped to form sTREM2, may be involved in the circRNA biogenesis from *TREM2*.

First, we designed divergent primers circTREM2_3–4 and circTREM2_4-5 (Figure 1A, Appendix A) in order to amplify circular, but not linear, RNA template that includes exon 4. Divergent primers are head-to-head oriented primers that target the exon or exons of interest. Next, we performed RT-PCR in RNA isolated from the human entorhinal cortex of controls. circTREM2_3-4 primers succeeded in amplifying a range of different molecular weight PCR products; in contrast, circTREM2_4-5 primers did not. After electrophoresis, we isolated a number of PCR products from agarose gels for Sanger sequencing analysis (Figure 1B, Appendix A). Following this approach, we identified three different sequences that included candidate back-splicing junctions between, on the one hand, exons 2 and 5 (circTREM2_1, circTREM2_2) and, on the other hand, intron 4 and exon 2 (circTREM2_3) (Figure 1C,D). As expected in circRNAs, the order of exons observed in these sequences (Figure 1C,D) did not correspond to the RNA linear form. Therefore, these sequences are assumed to belong to circRNAs and include circRNAs back-splicing junctions.

To make sure these sequences are actually part of circRNAs, we designed additional divergent primers but, in this case, overlapping other *TREM2* exons, namely circTREM2_2-4, circTREM2_2-3, and circTREM2_3-5 (Appendix A). After RT-PCR and Sanger sequencing, we observed that one of the three sequences described above, circTREM2_1 (Figure 1C,D), was amplified by using all three different primer sets (circTREM2_2-4, circTREM2_2-3 circTREM2_3-5). Thus, circTREM2_1 seems to correspond to a circRNA formed by exons 2, 3, 4, and 5, whose circRNA back-splicing junction is established between part of exon 2 and exon 5 (Figure 1A–D). Regarding the other two sequences (circTREM2_2 and circTREM2_3) (Figure 1A–D), they probably may be described as circRNAs, but were not amplified by all divergent primer sets.

To be noted, circTREM2_2 includes an SNP described as a risk factor for AD (rs143332484), and the 3′end of circTREM2_1 and circTREM2_3, when aligned to the human genome, is only 55 bp distant from rs143332484 and 100 bp away from rs75932628 (Appendix A), both genetic variants associated with AD.

Once we identified circRNAs derived from the *TREM2* gene in controls, we decided to perform the same approach in the entorhinal cortex of AD samples. In this case, we found similar results, and circTREM2_1 was amplified by all divergent primer sets, being the circTREM2 that we detected with the highest consistency.

### 2.2. circTREM2 in Human Microglia Cells

In the brain, the *TREM2* gene is almost exclusively expressed in microglia. Microglia involvement in AD is being extensively investigated, and although its role in the disease is still unclear, both protective and deleterious functions have been described [52]. Thus, we inquired whether any of the circTREM2 described in the previous section were expressed in microglia.

To address this question, we performed RT-PCR on RNA isolated from HMC3 cells, a human embryonic microglia cell line. Thus, we were able to confirm that both circTREM2_1 and circTREM2_3 were expressed in human microglia, although circTREM2_3, as in the case of the human entorhinal cortex, was not detected with all divergent primer sets (Appendix A).

### 2.3. circTREM2 Differential Expression in AD Entorhinal Cortex

In order to know whether a differential expression was observed between AD patients and controls in the entorhinal cortex, we carried out an RT-qPCR assay. For this experiment, we selected circTREM2_1 since it was the circTREM2 most consistently detected in our samples. To ensure that only circTREM2_1 was amplifying in the RT-qPCR assay, we designed a specific TaqMan-probe overlapping the back-splicing junction (Appendix A). Thus, we observed that circTREM2_1 was upregulated in AD samples (Fold-change (FC) = 2.53, *p*-value < 0.05) compared to controls (Figure 2A). However, after adjusting for age and gender, this difference was no longer statistically significant (FC = 1.7, *p*-value = 0.201). When examining the expression levels of circTREM2_1 considering the ABC score, we found that expression in the low ABC score cases was significantly increased compared to controls (*p*-value < 0.05) (Figure 2B). Again, after adjusting for age and gender, the association was no longer statistically significant.

### 2.4. TREM2 mRNA Expression Profiling and Differential Expression in AD Entorhinal Cortex

An increase in *TREM2* mRNA expression has been observed in the hippocampus and temporal cortex of AD patients [41,48,53]. Therefore, we wanted to study whether overall *TREM2* mRNA was also elevated in the entorhinal cortex of AD patients. By RT-qPCR, we observed that *TREM2* mRNA was upregulated in AD samples (FC = 2.47, *p*-value < 0.01) compared to controls (Figure 3A) and this difference remained statistically significant after adjusting for age and gender (FC = 2.33, *p*-value < 0.05).

In addition, we wanted to test whether the expression of any of the three *TREM2* transcripts underlies the overall higher *TREM2* mRNA expression observed in AD samples. We observed that the canonical transcript ENST00000373113 was the most highly expressed (*p*-value < 0.0001) (Figure 3B) and that the three *TREM2* transcript variants were upregulated in AD samples compared to controls (ENST00000373113: FC = 1.73, *p*-value < 0.05; ENST00000373122: FC = 2.19, *p*-value < 0.01; ENST00000338469: FC = 1.87, *p*-value < 0.05) (Figure 3C), However, after adjusting for age and gender, the difference between AD samples and controls was no longer statistically significant for any of them (ENST00000373113: FC = 1.62, *p*-value = 0.202; ENST00000373122: FC= 1.64, *p*-value = 0.162, ENST00000338469: FC = 1.47, *p*-value =0.302).

As a new variant of *TREM2* lacking exon 2 has recently been described in different areas of the human brain but not yet in the entorhinal cortex [45,46,47], we were interested to know if this isoform (*D2-TREM2*) is also expressed in the entorhinal cortex. Thus, we confirmed its expression in the entorhinal cortex and observed that it was upregulated in AD samples (FC = 2.02, *p*-value < 0.05), being the second transcript of *TREM2* most expressed; however, after adjusting for age and gender, the difference between AD samples and controls was no longer statistically significant (FC = 1.28, *p*-value = 0.50) (Figure 3B,C).

When analyzing the expression levels of *TREM2* mRNA according to the ABC score, we found that expression in the low and intermediate ABC score cases was significantly increased compared to controls (*p*-value < 0.05) (Appendix A). The association was also statistically significant after adjusting for age and gender.

According to the ABC score, the expression levels of the ENST00000373122 transcript were significantly increased in the intermediate ABC score cases compared to controls (*p*-value < 0.05) (Appendix A). Nevertheless, after adjusting for age and gender, the association was no longer statistically significant.

### 2.5. circTREM2, Overall TREM2, TREM2 Transcript Variants and Aβ Deposits

*TREM2* has been described as having an affinity for Aβ peptide, and this affinity is implicated in the activation of the signaling cascade for the clearance of Aβ deposits in AD [54,55,56]. Therefore, we wondered if circTREM2_1 expression showed any correlation with the Aβ burden. We observed a moderate negative correlation between the global average area of Aβ deposits in the entorhinal cortex and circTREM2_1 expression level among the AD cases (r = −0.434, *p*-value < 0.05) (Figure 4).

However, there was no correlation between expression levels of overall *TREM2* or *TREM2* transcript variants and the global average area of Aβ deposits in the entorhinal cortex. Neither there was a correlation between the expression level of circTREM2_1 and overall *TREM2* or *TREM2* transcript variants.

### 2.6. Prediction of miRNAs Binding Sites within circTREM2s

In order to assess the potential functional role of circTREM2, we used bioinformatics tools to predict miRNAs binding sites within the circRNA sequences. The miRDB software [57,58] was employed to find associated miRNAs. A total of 16 miRNAs were predicted to join with circTREM2_1, circTREM2_2, or circTREM2_3 (Table 1). Some of these miRNAs have been described to be involved in human pathological processes. For instance, miR-765 is implicated in osteogenesis [59] and has been reported as a tumor suppressor [60]. miR-939 seems to inhibit apoptosis and regulate angiogenesis through the nitric oxide signaling pathway, mainly in the context of coronary disease [61]. However, no circTREM2-related miRNAs have been associated with neurodegeneration or brain diseases so far.

## 3. Discussion

In this study, a *TREM2*-derived circRNA transcript (circTREM2_1) has been consistently identified in the human brain for the first time. We also show evidence for other circTREM2s in the human brain. In addition, we observed that circTREM2_1 is expressed in a cell model of human microglia. Additionally, overall *TREM2* mRNA expression was found to increase in the AD entorhinal cortex compared to controls, and we show the first report of the presence of the novel *D2-TREM2* isoform in the human entorhinal cortex. Interestingly, circTREM2_1, but not *TREM2* mRNA, expression levels negatively correlated with Aβ deposits in the entorhinal cortex of AD patients.

*TREM2* gene encodes a transmembrane glycoprotein receptor crucial in regulating microglia functions. Microglia are the innate immune cells in the brain and usually perform relevant surveillance tasks. In microglia, *TREM2* promotes phagocytosis of apoptotic cells and debris, misfolded proteins, and also decreases cytokine production, which is therefore of great relevance to maintaining brain homeostasis [62,63,64]. However, this homeostasis is altered in brain diseases, such as neurodegenerative conditions. Some microglial cells are then transformed into a particular state, known as disease-associated microglia (DAM) [65]. DAM was first described as a subset of brain immune cells characterized by a specific molecular signature identified by single-cell transcriptomics performed on brain immune cell samples of neurodegenerative diseases [65]. It has also been reported that DAM may early sense neuronal damage and play a protective role in neurodegeneration [66]. DAM is mainly detected in brain regions affected by AD and colocalizes with Aβ plaques [66]. Interestingly, microglia seem to depend on a molecular mechanism, which involves the Trem2-signaling pathway, to sense and respond to brain damage [66]. Thus, the *TREM2* gene is crucial to activating DAM in a neurodegenerative environment [65,67] since *TREM2* senses a wide range of damage signals and is able to maintain the DAM phenotype in response to these cues [66].

In this scenario, regulation of *TREM2* transcripts expression levels appears as a major node in preserving microglial functions in the brain. Epigenetic mechanisms have been previously described to regulate *TREM2* gene expression and/or to be altered in the AD context. For instance, DNA methylation is increased in the promoter region of *TREM2* in AD brain samples compared to controls in relevant brain regions such as the hippocampus [48] and prefrontal cortex [49,50,51]. Non-coding RNAs also participate in the epigenetic modulation of gene expression. miRNA-34a has been reported to target *TREM2* mRNA 3′ untranslated region (UTR) in human sporadic AD hippocampal CA1 and in primary microglia cell cultures, so it appears that miRNA-34a is involved in the down-regulation of *TREM2* [55,68,69]. In addition, the *TREM2* gene may be a source of epigenetic regulators aimed at self-regulation or modulation of other genes’ expression.

Recently, circRNAs have emerged as interesting molecules that deserve to be investigated as epigenetic regulators. circRNAs are a particular type of non-coding RNAs that are generated by a process known as back-splicing from a pre-mRNA. As a consequence of back-splicing, circRNAs form covalently close loops, which make them resistant to the action of RNase R and are, therefore, very stable molecules. circRNAs are thought to regulate alternative splicing and gene expression since back-splicing and linear splicing compete, and both molecular processes share the spliceosome machinery [13,26,70]. Indeed, among the functions of circRNAs are regulation of parental gene expression by competition with its linear counterparts [9,26], modulating translation of cognate mRNA [13], or occupying RNA binding sites in target genes [7]. circRNAs can also regulate protein expression and function by interacting with the protein or through protein complexes [71]. Another function of circRNAs is the ability to translate into proteins, as some of them have internal ribosomal entry site (IRES) within their sequences [32]. circRNAs may also act as miRNAs sponges disrupting their function as inhibitors of mRNAs [28,72].

The expression of circRNAs in the brain is high compared to other body tissues. Moreover, the expression of circRNAs in the brain changes throughout neuronal development, being in the adult stage when more circRNAs are detected; in other words, circRNAs accumulate with age. Their expression also changes according to brain region and cell subtypes. Research to date suggests that circRNAs may play important roles in synaptic plasticity and neuronal function [70,73,74].

Here, we describe a new circRNA transcript, circTREM2_1, arising from the microglial gene *TREM2* in the human brain and microglial cells. To our knowledge, no other *TREM2*-derived circRNA has been previously reported. circTREM2_1 includes almost all the exons of the *TREM2* gene (exons 2, 3, 4, and 5), including exon 4, which is most relevant for *TREM2* functioning since it encodes the transmembrane domain, along with most of cytoplasmatic and a small region of extracellular *Trem2* protein. In addition, exon 4 skipping seems to participate in the synthesis of sTREM2, which is significantly increased in the CSF of AD patients. As circTREM2_1 includes exon 4, upregulation of circTREM2_1 may favor the increased expression of sTREM2. However, whether a relationship between sTREM2 and circTREM2_1 is present should be unraveled by further investigations.

In our study, we observed the presence of circTREM2_1 in the entorhinal cortex of AD and control subjects, though no significant differences were revealed between both conditions. The entorhinal cortex is the most vulnerable region where AD neuropathological changes start in AD. DAM have been described as associated with brain regions where AD changes are first shown. Although differences in circTREM2_1 were no longer significant after adjusting for age and gender, it would be very interesting to investigate whether increases in circTREM2_1 expression levels are detrimental in the case of neurodegenerative diseases, in particular in AD, and the role that circTREM2 may have in impairing DAM maintenance. On the other hand, *TREM2* mRNA was found to increase in AD entorhinal samples and particularly between the low and intermediate ABC score cases compared to controls, which may perhaps be related to the progression of the disease.

In addition, a negative correlation was shown between circTREM2_1 expression levels and Aβ deposits in the entorhinal cortex. This is in contrast with a positive correlation previously described for *TREM2* mRNA levels and hippocampal Aβ burden [48], which we did not detect here. However, this result could be explained by a competition in the expression levels between the linear and circular forms of *TREM2*. Whether circTREM2_1 actually has a regulatory role in *TREM2* mRNA expression should be analyzed by additional experiments. On the other hand, a number of miRNAs have been predicted to be linked to circTREM2s. The function of most of them is unknown at this time and represents an area of special interest to be investigated.

## 4. Materials and Methods

### 4.1. Human Entorhinal Cortex Samples

Brain entorhinal cortex samples from 27 AD patients and 16 controls were provided by Navarrabiomed Brain Bank. After death, half brain specimens from donors were cryopreserved at −80 °C. Neuropathological examination was completed following the usual recommendations [75] and according to the updated National Institute on Aging-Alzheimer’s Association (NIA-AA) guidelines [76].

Assessment of Aβ deposition was carried out by immunohistochemical staining of paraffin-embedded sections (3–5 μm thick) with a mouse monoclonal (S6 F/3D) anti-Aβ antibody (Leica Biosystems Newcastle Ltd., Newcastle upon Tyne, UK). Evaluation of neurofibrillary pathology was performed with a mouse monoclonal antibody anti-human PHF-TAU, clone AT-8 (Tau AT8) (Innogenetics, Gent, Belgium), which identifies hyperphosphorylated tau (p-tau) [77]. The reaction product was visualized using an automated slide immunostainer (Leica Bond Max) with Bond Polymer Refine Detection (Leica Biosystems, Newcastle Ltd., Newcastle upon Tyne, UK). Other protein deposits, such as synuclein deposits, were ruled out by a monoclonal antibody against α-synuclein (NCL-L-ASYN; Leica Biosystems, Wetzlar, Germany). The staging of AD was performed by using the ABC score according to the updated NIA-AA guidelines [76]. ABC score combines histopathologic assessments of Aβ deposits determined by the method of Thal (A) [76], staging of neurofibrillary tangles by Braak and Braak classification (B) [77], and scoring of neuritic plaques by the method of CERAD (Consortium to Establish a Registry for Alzheimer’s Disease) (C) [78] to characterize AD neuropathological changes. Thus, the ABC score shows three levels of AD neuropathological severity: low, intermediate, and high. A summary of the characteristics of subjects considered in this study is shown in Appendix A.

### 4.2. Microglial Cell Culture

Human embryonic microglia clone 3 (HMC3, ATCC^®^ CRL3304™) cells were cultured in minimum essential medium (MEM) Eagle-EBSS with NEAA without L-Glutamine (Lonza, Cologne, Germany) supplemented with 10% fetal bovine serum (Sigma-Aldrich, Saint Louis, MO, USA), 1% penicillin-streptomycin (Gibco™ Thermo Fisher Scientific, Waltham, MA, USA,), and 1% L-glutamine 200mM (Gibco™, Thermo Fisher Scientific, Waltham, MA, USA) at 37 °C and in an atmosphere of 5% CO_2_.

### 4.3. RNA Isolation and Reverse Transcription–Polymerase Chain Reaction (RT-PCR)

Total RNA was isolated from cells and entorhinal cortex samples with miRNeasy mini-Kit (QIAGEN, Redwood City, CA, USA) following the manufacturer’s instructions. The concentration and purity of RNA were both evaluated with a NanoDrop spectrophotometer. Complementary DNA (cDNA) was reverse transcribed from 500 ng total RNA with SuperScript^®^ III First-Strand Synthesis Reverse Transcriptase (Invitrogen, Carlsbad, CA, USA) after priming with random primers. RT-PCR was performed by using GoTaq^®^ DNA polymerase (Promega, 2800 Woods Hollow Road, Madison, WI, USA) in an Applied Biosystems™ Veriti™ Thermal Cycler, 96-Well (Applied Biosystems, Foster City, CA, USA). PCR conditions were as follows: denaturation at 95 °C for 20 s, extension at 72 °C for 30 s, and annealing temperatures at 60 °C for 40 s, and cycles used were 40. Primer3 software was used for divergent primers design (Appendix A).

### 4.4. Candidate Band Isolation and Sanger Sequencing

Candidate bands were selected after 1.8% agarose gel electrophoresis of RT-PCR products. Bands purification was made with Wizard^®^ SV Gel and PCR Clean-Up System (Promega, 2800 Woods Hollow Road,·Madison, WI, USA). Next, Sanger sequencing was performed, and UCSC (University of California Santa Cruz) Genome Browser software was used for the sequence alignment [42].

### 4.5. Real-Time Quantitative PCR (RT-qPCR) Assay

Total RNA was isolated from the entorhinal cortex with RNeasy Lipid Tissue mini-Kit (QIAGEN, Redwood City, CA, USA) following the manufacturer’s instructions. Genomic DNA was removed with recombinant DNase (TURBO DNA-free™ Kit, Ambion, Austin, TX, USA). The concentration and purity of RNA were both evaluated with a NanoDrop spectrophotometer. cDNA was reverse transcribed from 500 ng total RNA with SuperScript^®^ III First-Strand Synthesis Reverse Transcriptase (Invitrogen, Carlsbad, CA, USA) after priming with random primers. RT-qPCR reactions were performed in triplicate with TaqMan Master Mix: Premix Ex Taq (TaKaRa, Otsu, Japan) in a QuantStudio 12K Flex Real-Time PCR System (Applied Biosystems, Foster City, CA, USA). Sequences of primer pair and probe were designed using the Real-Time PCR tool (IDT, Coralville, IA, USA) (Additional Table 1). The relative expression level of mRNA in a particular sample was calculated as previously described [79], and the ACTB gene was used as the reference gene to normalize expression values.

RT-qPCR reactions for overall and transcript-specific *TREM2* mRNA quantification were performed in triplicate with Power SYBR Green PCR Master Mix (Invitrogen, Carlsbad, CA, USA) in a QuantStudio 12K Flex Real-Time PCR System (Applied Biosystems, Foster City, CA, USA) and repeated twice within independent cDNA sets. Sequences of primer pairs were designed using the Real-Time PCR tool (IDT, Coralville, IA, USA) and Primer3 software and *D2-TREM2* primers described in Shaw B.C. et al. [47] were included (Appendix A). The relative expression level of mRNA in a particular sample was calculated as previously described [80], and the geometric mean of *GAPDH* and *ACTB* genes was used as a reference to normalize expression values.

### 4.6. Quantitative Assessment of Aβ Deposits in Brain Tissues

In order to quantitatively assess the Aβ burden for further statistical analysis, we applied a method to quantify protein deposits. This method generates a numeric measurement that represents the extent of Aβ deposition. Sections of the entorhinal cortex were examined after performing immunostaining with an anti Aβ antibody as described above in Human Entorhinal Cortex Samples. Three pictures were obtained for each immunostained section by using an Olympus BX51 microscope at ×10 magnification power. The focal deposit of Aβ, as described by Braak and Braak (neuritic, immature, and compact plaque) [77], was manually determined and was further edited and analyzed with the ImageJ software. Then, Aβ plaque count, referred to as amyloid plaque score (APS), and total area of Aβ deposition was automatically measured by ImageJ and averaged for each section.

### 4.7. Statistical Analysis

Statistical analysis was performed with SPSS 25.0 (IBM, Inc., Amonk, NY, USA). Before performing differential analysis, we checked whether continuous variables follow a normal distribution, as per one-sample Kolgomorov–Smirnov test and the normal quantile–quantile (QQ) plots. The values of expression levels were transformed by using log10(X) to meet the conditions of normality, then Student’s *t*-test was used to analyze differences in the expression levels of circTREM2_1 and *TREM2* mRNA. One-way analysis of variance (ANOVA) and post hoc Bonferroni test was used to analyze differences in the expression levels of circTREM2_1 and *TREM2* mRNA among ABC stage groups. A general linear model univariate analysis was used to adjust for gender and age. Pearson test was used to assess the correlation between continuous variables. Significance level was set at *p*-value < 0.05. GraphPad Prism version 6.00 for Windows (GraphPad Software, La Jolla, CA, USA) was used to draw graphs.

## 5. Conclusions

In conclusion, these results show that a *TREM2*-derived circRNA (circTREM2_1) is consistently expressed in the entorhinal cortex of both AD and control brains. In addition, cell culture data suggest that microglia may be a source of this circRNA in the human brain. Interestingly, *TREM2* mRNA levels were found elevated in entorhinal samples of AD patients, specifically between those showing low and intermediate ABC scores and controls. It is worth mentioning that the novel *D2-TREM2* isoform is also expressed in the entorhinal cortex. *TREM2* is a pivotal gene in microglia functioning and shaping DAM, and thus, dissecting the expression profile of the *TREM2* gene, including its non-coding RNA isoforms, adds significant information to the process of microglia activation and neuroinflammation in neurodegenerative diseases. The role of circRNAs in the brain and, in particular, in neurodegeneration still needs to be clarified.

At the present time, it is worth remembering that many circRNAs have been described or predicted only on the basis of in silico bioinformatic studies. However, biological confirmation of their expression in tissues of interest, as is the case in our study, is very valuable in revealing their expression. In the case of circTREM2s, their presence had not been reported either in databases or in biological studies, so this article has this added value.

Although no significant differences in circTREM2_1 or linear *TREM2* isoforms expression levels were observed between AD patients and controls in our sample set, this could be due to the reduced sample size to detect these changes. Therefore, measurement of these variants and, particularly, circTREM2_1 expression levels in larger cohorts would help to elucidate whether differences actually exist. Interestingly, changes in circTREM2_1 expression levels could have transcriptional consequences on other genes regulated by miRNAs for which circTREM2_1 would act as a sponge. In any case, further investigations are needed to unravel the potential detrimental role of circTREM2_1 in the pathogenesis of AD.

## Figures and Tables

**Figure 1 ijms-23-07682-f001:**
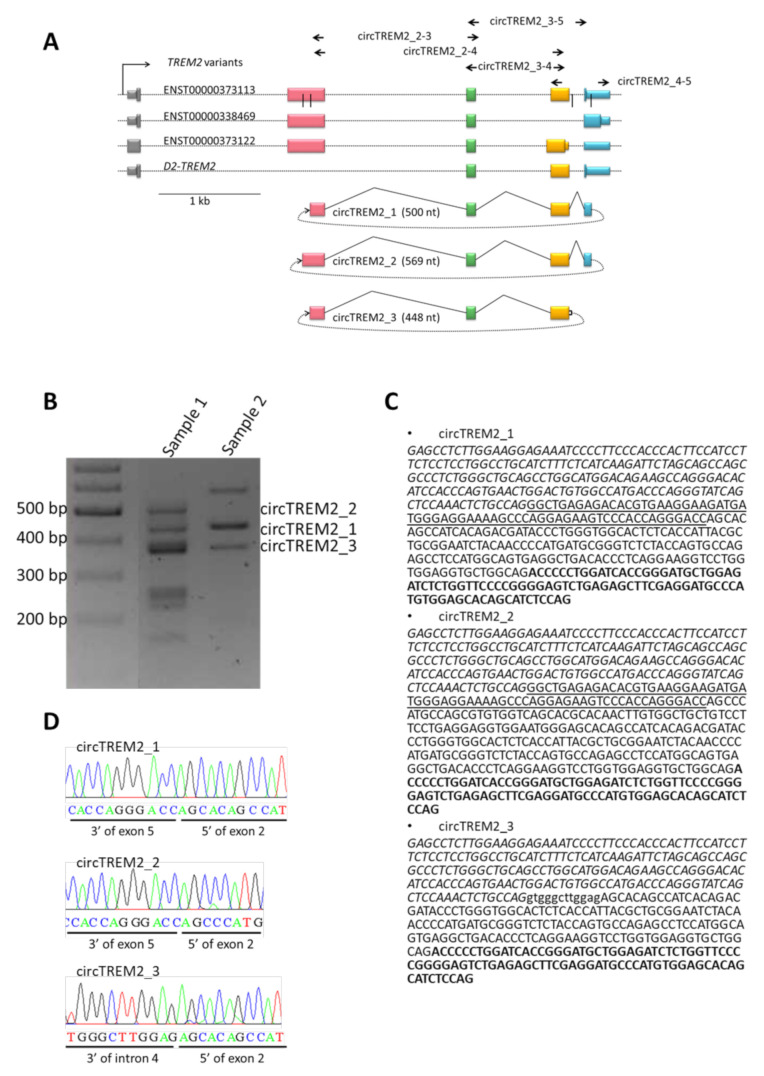
*TREM2* and circTREM2 representation. (**A**) Gene map of linear and circular *TREM2* RNA. Boxes represent exons; black vertical lines are back-splicing junction sites. Dotted lines indicated the back splicing of exons. Arrows represent set of primers used. (**B**) Representative figure of RT-PCR products of circTREM2_3-4 primer set in agarose gel. (**C**) Sequences of circTREM2s, circRNA part underlined correspond to exon 5, regular letter to exon 2, italic letter to exon 4, letter in bold to exon 3, and lower case to intron. (**D**) Electropherograms show the sequence of the junction of each circTREM2, confirming the fusion between 3’ of exon 5 and 5’ of exon 2 in the case of circTREM2_1 and circTREM2_2 and the fusion between 3’ of intron 4 and 5’ of exon 2.

**Figure 2 ijms-23-07682-f002:**
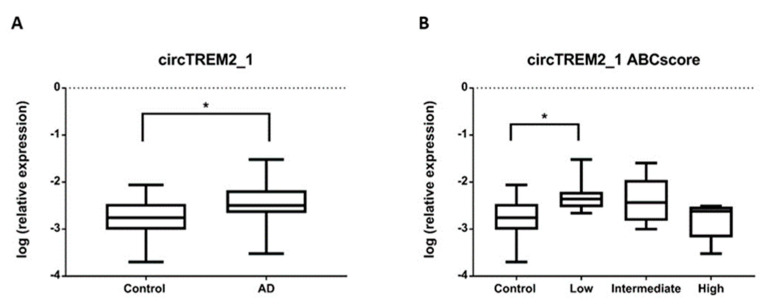
circTREM2_1 expression levels relative to *ACTB* housekeeping gene expression. (**A**) The graph shows a significant increase in circTREM2_1 levels in AD entorhinal cortex samples compared to control entorhinal cortex samples. (**B**) circTREM2_1 expression decreased across AD stages, as shown when circTREM2_1 expression levels are sorted by ABC score. Vertical lines represent the standard error of the mean. * *p*-value < 0.05.

**Figure 3 ijms-23-07682-f003:**
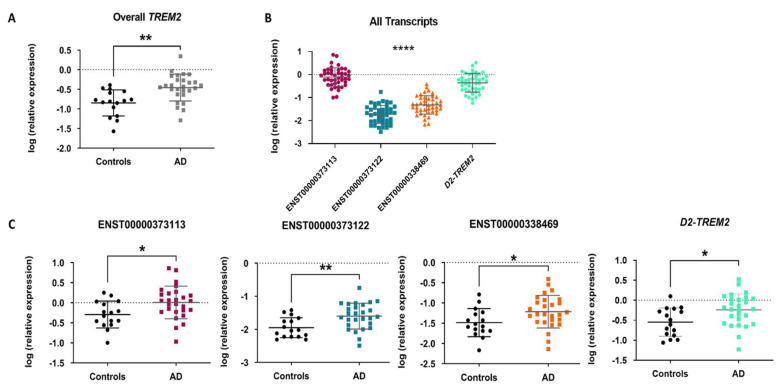
Overall and transcript-specific *TREM2* expression levels. (**A**) Dot-plot represents the expression of *TREM2* mRNA in AD samples and controls. (**B**) Dot-plot shows the expression of *TREM2* transcripts variants in entorhinal cortex. Vertical lines represent the SE. (**C**) Dot-plot represents the expression level of each *TREM2* transcript variant in entorhinal cortex of AD patients when compared to controls. (**C**) * *p*-value < 0.05, ** *p*-value < 0.01, and **** *p*-value < 0.0001.

**Figure 4 ijms-23-07682-f004:**
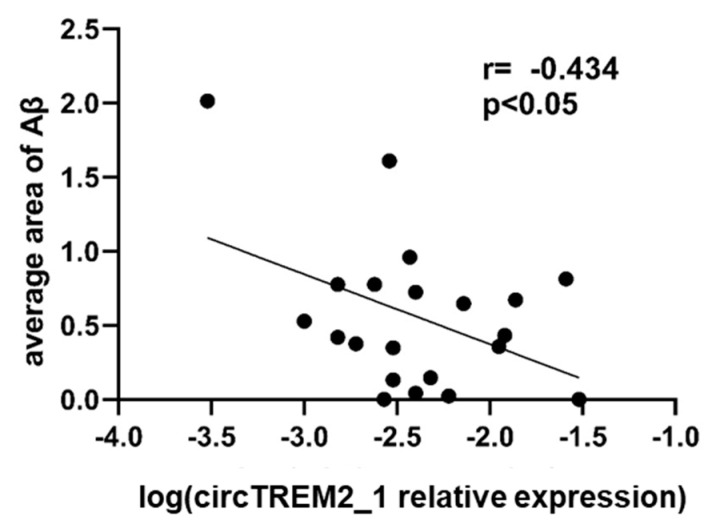
circTREM2 and Aβ deposits correlation. Dispersion diagram showing negative correlation between global average area of Aβ deposits and circTREM2_1 expression levels.

**Table 1 ijms-23-07682-t001:** miRNAs predicted with miRBD software to bind circTREM2s.

miRNAs Predicted to Join	circTREM2_1	circTREM2_2	circTREM2_3
hsa-miR-765	Yes	Yes	No
hsa-miR-11181-3p	Yes	Yes	No
hsa-miR-6890-3p	Yes	Yes	Yes
hsa-miR-6766-5p	Yes	Yes	No
hsa-miR-6756-5p	Yes	Yes	No
hsa-miR-653-3p	Yes	Yes	Yes
hsa-miR-6770-5p	Yes	Yes	No
hsa-miR-6131	Yes	No	No
hsa-miR-6762-3p	Yes	Yes	No
hsa-miR-4783-3p	Yes	Yes	Yes
hsa-miR-2392	Yes	Yes	No
hsa-miR-4483	Yes	Yes	No
hsa-miR-6745	Yes	Yes	No
hsa-miR-6749-3p	No	No	Yes
hsa-miR-939-3p	No	No	Yes
hsa-miR-3657	No	No	Yes

## Data Availability

The datasets used and/or analyzed during the current study are available from the corresponding author on reasonable request.

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
