# Peer review of "Profile of TREM2-Derived circRNA and mRNA Variants in the Entorhinal Cortex of Alzheimer’s Disease Patients"

_ijms, 2022, doi:10.3390/ijms23147682_

Round 1
Reviewer 1 Report
The authors responded to all the reviewer comments in the revised manuscript.
Author Response
We are very grateful of the reviewers’ effort in revising the manuscript and their constructive suggestions.
Reviewer 2 Report
The authors have improved the results section by adding new figures. Moreover, a conclusion section has been added.
Author Response

(The authors gave the same response as above.)

Reviewer 3 Report
Here the authors identified circular RNAs derived from TREM RNA. The work is intriguing; however, the following major issues dampened my enthusiasm for this manuscript.
· Fig. 2: The fact that the levels of circTREM2_1 are not different (after adjusting for age and gender) between AD and CTL is a bit disappointing. Are the changes in circTREM2_1 simply due to an age-dependent process, independent of AD pathology? Are the levels in TREM2_1 different between male and female controls? The authors must answer these two questions to better understand if this circular RNA might be involved in AD pathogenesis.
· The point above makes the title of the manuscript misleading as the authors did not find any changes in the expression of circTREM RNAs after adjusting their data for the proper controls.
· The data in Fig. 4 are interesting. However, one wonders if the driver of the correlation is age and not Aβ
Round 2
Reviewer 3 Report
The authors have addressed all of my comments.
Author Response
We thank Reviewer 3 for their effort and feedback.
This manuscript is a resubmission of an earlier submission. The following is a list of the peer review reports and author responses from that submission.
Round 1
Reviewer 1 Report
The manuscript by Urdánoz-Casado et al. is interesting and well written. The authors report for the first time the identification of TREM2-derived circRNA transcript (circTREM2_1) in the human brain. They also demonstrated the presence of other circTREM2s in the human brain. In addition, they show that circTREM2_1 is expressed in a cell model of human microglia. Interestingly, circTREM2_1 expression levels negatively correlated with amyloid-beta deposits in the entorhinal cortex of AD patients. I think the paper is publication-worthy in IJMS nonetheless a minor revision is required. Results description and figures should be improved. In particular, figures regarding 2.2, 2.3, and 2.4 sections are missing in the main text, I think they should be included in the paper, in order to make the paper more readers friendly and, to better discuss the obtained results.
Author Response
1.- The manuscript by Urdánoz-Casado et al. is interesting and well written. The authors report for the first time the identification of TREM2-derived circRNA transcript (circTREM2_1) in the human brain. They also demonstrated the presence of other circTREM2s in the human brain. In addition, they show that circTREM2_1 is expressed in a cell model of human microglia. Interestingly, circTREM2_1 expression levels negatively correlated with amyloid-beta deposits in the entorhinal cortex of AD patients. I think the paper is publication-worthy in IJMS nonetheless a minor revision is required. Results description and figures should be improved. In particular, figures regarding 2.2, 2.3, and 2.4 sections are missing in the main text, I think they should be included in the paper, in order to make the paper more readers friendly and, to better discuss the obtained results.
We thank reviewer #1 for the positive comments. As suggested, figures have been added in 2.2, 2.3 and 2.4 sections showing Additional Figure 4: RT-PCR products of circTREM2_2-3 primers set from HMC3 cells in agarose gel; Figure 2: circTREM2_1 expression levels relative to ACTB housekeeping gene expression and Figure 3: circTREM2 and Aβ deposits correlation.
Reviewer 2 Report
In this manuscript, the authors studied the TREM2-derived circRNA, a novel class of noncoding RNAs characterized by a covalent and stable closed loop structure of Alzheimer’s disease patients. Overall, the manuscript is well-conducted and discussed. But there are some issues to be addressed regarding the publication.
1. In the manuscript, there is no section devoted to the conclusion. Please include and summarize this section by presenting transcriptional consequences as well as a brief author's perspective.
2. For the results of quantitative assessment and statistical analysis, it would be better if these results could be expressed by the graphs associated with each table, so that the readers could easily access them.
3. English should be corrected carefully to remove grammatical errors in the manuscript.
Author Response
2.-In this manuscript, the authors studied the TREM2-derived circRNA, a novel class of noncoding RNAs characterized by a covalent and stable closed loop structure of Alzheimer’s disease patients. Overall, the manuscript is well-conducted and discussed. But there are some issues to be addressed regarding the publication.
In the manuscript, there is no section devoted to the conclusion. Please include and summarize this section by presenting transcriptional consequences as well as a brief author's perspective.
We would like to thank the reviewer to point this issue to our attention. Now, a conclusion section has been added to the main text (lines 293-318; highlighted in blue font).
- For the results of quantitative assessment and statistical analysis, it would be better if these results could be expressed by the graphs associated with each table, so that the readers could easily access them.
Thanks for the advice. In the Results sections 2.3 and 2.4, graphs have been added to improve the legibility of the manuscript.
English should be corrected carefully to remove grammatical errors in the manuscript.
Grammar has been corrected as thoroughly as we have been able to.
Reviewer 3 Report
The manuscript writing is not good. Methods and results have fundamental errors. Data is very superficial type. Discussion section is not well correlated with the exiting articles. Overall, manuscript is not suitable to
publish in this high impact journal.
Author Response
The manuscript writing is not good. Methods and results have fundamental errors. Data is very superficial type. Discussion section is not well correlated with the exiting articles. Overall, manuscript is not suitable to publish in this high impact journal.
We regret that the reviewer evaluated the manuscript in this way. In any case, we would be willing to review and discuss in detail each of the comments expressed here in a generic way.
In our favor, we would like to point out that we have previously published work of similar quality in this journal (3 articles, including one paper on circRNAs in Alzheimer's disease).
Round 2
Reviewer 3 Report
Data is insufficient for publication.